# Margin between success and failure of PDA stenting for duct-dependent pulmonary circulation

**Hala Mounir Agha**[1]*, **Osama Abd -El Aziz**[1], **Ola Kamel**[1], **Sahar S. Sheta**[1], **Amal El-Sisi**[1], **Sonia El-Saiedi**[1], **Aya Fatouh**[1], **Amira Esmat**[1], **Gaser Abdelmohsen**[1], **Baher Hanna**[1], **Mai Hussien**[2], **Rodina Sobhy**[1]

**1** Department of Pediatrics, Pediatric Cardiology Division, Specialized Pediatric Hospital, Cairo University, Cairo, Egypt, **2** Pediatric Department, General Organization of Teaching Hospitals and Institues, Cairo, Egypt

* halaazza@gmail.com

**Data Availability Statement:** The minimal data set underlying the results described in the manuscript can be found in https://doi.org/10.5061/dryad.ht76hdrdk.

## Abstract

### Objectives

Percutaneous patent ductus arteriosus (PDA) stenting is a therapeutic modality in patients with duct-dependent pulmonary circulation with reported success rates from 80–100%. The current study aims to assess the outcome and the indicators of success for PDA stenting in different ductal morphologies using various approaches.

### Methods

A prospective cohort study from a single tertiary center presented from January 2018 to December 2019 that included 96 consecutive infants with ductal-dependent pulmonary circulation and palliated with PDA stenting. Patients were divided according to PDA origin into 4 groups: Group 1: PDA from proximal descending aorta, Group 2: from undersurface of aortic arch, Group 3: opposite the subclavian artery, Group 4: opposite the innominate/brachiocephalic artery.

### Results

The median age of patients was 22 days and median weight was 3 kg. The procedure was successful in 78 patients (81.25%). PDA was tortuous in 70 out of 96 patients. Femoral artery was the preferred approach in Group 1 (63/67), while axillary artery access was preferred in the other groups (6/11 in Group 2, 11/17 in Group 3, 1/1 in Group 4, P <0.0001). The main cause of procedural failure was inadequate parked coronary wire inside one of the branch of pulmonary arteries (14 cases; 77.7%), while 2 cases (11.1%) were complicated by acute stent thrombosis, and another 2 cases with stent dislodgment. Other procedural complications comprised femoral artery thrombosis in 7 cases (7.2%). Patients with straight PDA, younger age at procedure and who had larger PDA at pulmonary end had higher odds for success (OR = 8.01, 2.94, 7.40, CI = 1.011–63.68, 0.960–0.99, 1.172–7.40, respectively, P = 0.048, 0.031, 0.022 respectively).

**Funding:** The author(s) received no specific funding for this work.

**Competing interests:** The authors have declared that no competing interests exist.

## Conclusions

The approach for PDA stenting and hence the outcome is markedly determined by the PDA origin and morphology. Patients with straight PDA, younger age at procedure and those who had relatively larger PDA at the pulmonary end had better opportunity for successful procedure.

## Introduction

Cyanotic congenital heart diseases with duct-dependent pulmonary circulation are considered neonatal emergencies that need immediate intervention. PDA stenting has emerged as a preferred method for palliation: to maintain adequate oxygen saturation and to promote pulmonary vascular growth. Newer techniques and tools along with improving operators skills have all improved the outcomes in the current era [1–8]. The variations in ductal origin and morphology may be an obstacle for the operator that necessitate alternative approaches other than the femoral artery [9–11]. Little is known about the outcome after PDA stenting in neonates and infants with duct-dependent cyanotic congenital heart disease using different approaches. This study aimed to evaluate the outcome and the indicators of the success for the ductal stenting in various PDA morphology and origin using different vascular approaches.

## Patients and methods

A prospective cohort study that included all patients with duct-dependent pulmonary circulation treated by ductal stenting at Cairo University Specialized Pediatric Hospital from January 2018 till December 2019. Patients with untreated sepsis or coagulopathies were excluded. The study was approved from the Faculty of medicine, Cairo University ethical committee. Informed consents were obtained from the patient's legal guardian for the procedure and the utilization of data. High risk written consent was taken from the guardian of each patient before the procedure according to the institutional rules. Patients and the public were not involved in the study design, analysis, interpretation and writing of the study.

*Data* were collected in an excel sheet including clinical data (age, body weight and oxygen saturation); *echocardiographic data* (cardiac anatomical diagnosis, PDA morphology (origin, shape and size); catheterization *data* (vascular access, description of the PDA [origin, morphology and diameter at both pulmonary and aortic ends], size of the pulmonary arteries, diameter, length and the type of the stent used, fluoroscopy time, radiation time, related complications and added interventions in the same setting). Echocardiographic studies were performed using a 6 MHZ probe of GE Vivid E 5 echo machine (Vingmed Ultrasound AS, Horten, Norway) and cardiac catheterizations were performed using a Philips Allura Xper FD10 catheterization laboratory (Philips North America Corporation, Cambridge MA, USA). *Procedural details*: all procedures were performed under general anaesthesia, and femoral arterial 4F accesses were obtained followed by IV unfractionated heparin 100 IU/kg bolus plus prophylactic antibiotic. Aortograms in the frontal and lateral projections using pig-tail catheters to identify the ductal origin, morphology, length and diameters at the pulmonary end, and the pulmonary arterial branch anatomy. According to the ductal origin, the patients were divided into: group 1 the PDA arises from the proximal descending aorta; group 2 from the undersurface of aortic arch; group 3 opposite to the left subclavian artery; and group 4 opposite to the innominate/brachiocephalic artery [Fig 1A–1H]. Based on the angiogram, the approach

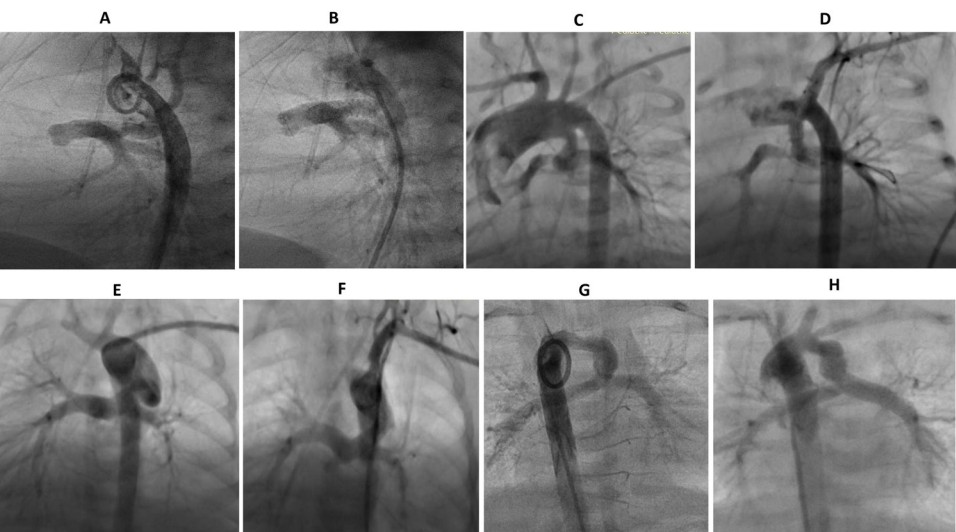

**Fig 1. Angiography done during cardiac catheterization showing various PDA origins and morphology before and after ductal stenting. A-B:** Group 1 with PDA arises from the proximal descending aorta, **C-D:** Group 2 with PDA arises from the undersurface of aortic arch, **E-F:** Group 3 with PDA opposite the left subclavian artery, **G-H:** Group 4 with PDA opposite the innominate/Brachiocephalic trunk.

was decided, i.e. whether to proceed via the femoral arterial access or to convert to an alternate axillary, carotid or femoral vein access. The ductus was accessed via a cut pig-tail or Judkins right [JR] catheter, and crossed with 0.014" percutaneous transluminal coronary angioplasty [PTCA] wire [middle support weight guide wire, Boston Scientific, USA] that was parked in one of the branch pulmonary arteries distally. Stents used were pre-mounted bare metal stents [architect, Life Vascular Device Biotech, Barcelona, Spain] or drug-eluting coronary stents [Monorail, Everolimus-Eluting Platinum Chromium, Boston Scientific, USA]. Stent diameter was chosen with respect to the weight of the patient, minimal PDA diameter, and branch pulmonary arteries. Length of stent was chosen to cover the entire length of the duct with proximal end into aorta and distal end into branch PA. In case of branch pulmonary stenosis, the stents aimed to reach distal to the stenosis. The selected coronary stent was tracked over PTCA wire and centered across the PDA. After confirming the position, stent was deployed using the initial nominal pressure then redilated for better apposition. Balloon was then removed after deployment keeping the wire in situ. Repeat angiograms were done to check the stent position and flow to branch pulmonary arteries. After achieving satisfactory results, wire was removed. Post-procedure details: All the patients were transferred to ICU and heparin infusion was maintained for at least for 24–48 hours with monitoring partial prothrombin time to be 2–3 times of normal. Acetylsalicylic acid 5mg/kg/day was initiated when oral feeding was established and overlapped for 12 hours, and continued until undergoing second-stage palliation or repair. Procedural success of PDA stenting was defined as the implantation of a well-seated stent, good flow for both pulmonary arteries and stable SpO2 above 75% and a hemodynamically-stable patient.

## Follow-up

Patients were followed during visits by pulse oximetry to assess oxygen saturation, by echocardiography to evaluate stent patency, size of the pulmonary arteries and neointimal proliferation within the stent. Multidetector computed tomography [MDCT] was performed whenever

oxygen saturation falls beneath 75%, or prior to planned surgical intervention for preoperative evaluation of the stent patency and the pulmonary arterial growth.

## Statistical analysis

Statistical analysis was performed using SPSS Version 20 software [IBM, Armonk, New York] and Med Calc for Windows, version 15.0 [Med Calc Software, Ostend, Belgium]. Numerical data were expressed as median and interquartile range [25th -75th percentiles] while categorical data were expressed as numbers or numbers and percentages. Comparisons between groups were calculated using non-parametric Kruskal Wallis, Mann-Whitney and Wilcoxon signed-rank tests for numerical data, while χ2 tests were used for comparison between groups for categorical data. Logistic regression was performed for evaluation of outcome predictors. Statistical significance was considered if P-values <0.05.

## Results

We attempted a total of 96 cases for PDA stenting during the study period: their median age was 22 days [ranging from 2 to 120 days], median body weight was 3.05 kg [2.2 to 3.5 kg], and mean length was 50 cm [47 to 52 cm] and 66 [68.8%] were males [Table 1]. Twelve (12.5% of cases) were smaller than 2.5 kg). Univentricular repair was planned for 55 cases (57.3%) and biventricular repair for 41 cases [42.7%]. Table 1 lists the different cardiac anomalies encountered and the additional procedures that were performed in 27 patients (28.1%). Based on ductal origin, group 1 comprised 67 patients (69.8%); group 2 was 11 patients (11.4%); group 3 was 17 patients (17.7%); and group 4 had only 1 patient (1%) that was not included in the statistical analysis being a single case. Ductal morphologies were either tortuous (70 patients, 72.9%) or straight (26 patients, 27.1%). Based on ductal origin and morphology, the approach to ductal stenting was the femoral artery in 72 patients (75%); axillary artery in 19 patients (19.8%); carotid artery in 3 patients (3.1%); and femoral vein in 2 patients (2.1%). It is of note that 2 cases that were initiated via the femoral artery approach then shifted to the axillary route.

Successful stenting was achieved in 78 cases (81.25%) of these, 53 patients had a tortuous PDA (67.9%) and in 25 patients the duct was straight (32.1%). Out of the 78 successful cases, 54 were of group 1 (69.2%); 9 patients of group 2 (11.5%); 14 patients of group 3 (17.9%); and the one patient of group 4 (1.3%). The approach was femoral artery in 60 patients (67.9%), axillary artery in 13 patients (16.6%); carotid artery in 3 patients (3.8%); and femoral vein in 2 patients (2.6%). Within the 18 failed cases (18.75%), 17 were of the tortuous morphology (94.4%); 12 were of group 1 (72.2%); 2 were of group 2 (11.1%); 3 of group 3 (16.7%); and none of group 4. The approach was femoral artery in 12 cases (66.7%); and axillary artery in 6 cases (33.3%). The main cause of procedural failure was inadequate parking of the coronary wire in a distal pulmonary arterial branch in 14 cases (77.7%), while 2 cases (11.1%) were complicated by acute stent thrombosis, and another 2 cases with stent dislodgment. Other procedural complications comprised femoral artery thrombosis in 7 cases (7.2%), No patients died on table or within 6 hours of the procedure.

### Analysis of results according to ductal origin

**Group 1: PDA originating from proximal descending aorta (n = 67).** Successful stenting was achieved in 54 patients (80.6%), while the other 13 cases were referred for surgical shunt. Tortuous ductus was encountered in 48 patients (71.6%). The most preferred access route for PDA stenting in group I was the femoral artery (n = 63, 94%). Regarding procedural complications: two patients experienced acute stent embolization, one patient developed

**Table 1. Characteristics of patients' groups categorized based on PDA origin.**

| | Group 1 (n = 67) | Group 2 (n = 11) | Group 3 (n = 17) | Group 4 (n = 1) | P- value |
|---|---|---|---|---|---|
| **Age, days** | 24(11–42) | 21(13–46) | 19(8–32) | 11 | 0.566 |
| **Weight, Kg** | 3(2.7–3.5) | 3(2.5–3.3) | 3.3(3–3.7) | 3.2 | 0.415 |
| **Diagnosis, n** | | | | | |
| PA-IVS | 24 | 1 | 1 | | |
| PA-VSD | 10 | 4 | 7 | 1 | |
| PA-TA | 15 | | 2 | | |
| DORV-PA | 3 | 1 | | | |
| Heterotaxy syndrome | 7 | 3 | 2 | | |
| Critical PS | 3 | | | | |
| SV-PA | 4 | 2 | 5 | | |
| Ebstein Anomaly | 1 | | | | |
| **Vascular access, n** | | | | | <0.0001* |
| Femoral artery | 63 | 4 | 5 | | |
| Axillary artery | 1 | 6 | 11 | 1 | |
| Carotid artery | 1 | 1 | 1 | | |
| Femoral vein | 2 | | | | |
| **Using 2 stents, n** | 1 | | 1 | | |
| **Stent type, n** | | | | | |
| Drug eluting | 10 | 2 | 5 | | |
| Bare metal | 46 | 8 | 11 | 1 | |
| **Stent diameter, mm** | 3.5(3–4.5) | 3.5(3.5–4) | 3.5(3.5–4) | 4 | 0.621 |
| **Stent length, mm** | 18(12–28) | 18(14–24) | 24(14–24) | 24 | 0.022* |
| **PDA shape(tortuous/straight), n** | 48/19 | 9/2 | 13/4 | 0/1 | 0.348 |
| **Additional procedures, n** | | | | | |
| Pulmonary valvuloplasty | 4 | 1 | | | |
| Atrial septostomy | 13 | 1 | | | |
| Radiofrequency perforation | 4 | | | | |
| Aortic valvuloplasty | 1 | | | | |
| **Radiation dose, (Gycm²)** | 39.5(19.8–83.2) | 45(24–68) | 43(13.2–79) | 16 | 0.648 |
| **Fluoroscopy time, (min)** | 23.2(11.5–37.1) | 14(10.8–30.0) | 24.5(13.7–30.9) | 15.5 | 0.753 |
| **Complications, n** | | | | | |
| Stent Embolization | 2 | | | | |
| Femoral artery thrombosis | 6 | | 1 | | |
| Cardiac arrest/CPR | 1 | | | | |
| Bradycardia | 1 | | | | |
| Immediate stent thrombosis | | 1 | 1 | | |
| Protrusion to RPA or LPA | 2 | | 4 | | |
| Uncovered aortic end | 4 | 1 | 1 | | |
| **Success/Failure, n†** | 54/13 | 9/2 | 14/3 | 1/0 | 0.966 |
| **Discharge/In hospital mortality, n‡** | 40/14 | 8/1 | 9/5 | 1/0 | 0.628 |
| **Procedure related Mortality, n** | 3 | 1 | 1 | | |
| **Hospital stay, days** | 5(1–150) | 5(1–28) | 6(1–60) | 6 | 0.984 |
| **Reintervention (follow up), n** | 3 (2 BAS, 1 PDA stent | 2 (1stent dilatation, 1additional stent) | | | |

PA-IVS: Pulmonary atresia with intact ventricular septum, PA-VSD: Pulmonary atresia with ventricular septal defect, PA-TA: Pulmonary atresia with tricuspid atresia, DORV-PA: Double outlet right ventricle with pulmonary atresia, SV-PA: Single ventricle with pulmonary atresia, PS: Pulmonary stenosis, PDA: Patent ductus arteriosus, CPR: Cardiopulmonary resuscitation, RPA: Right pulmonary artery, LPA: Left pulmonary artery. BAS: Balloon atrial septostomy

†: Failed cases were referred for surgical shunt

‡: For the success group

*: Statistically significant.

cardiac arrest and successfully resuscitated, and one patient developed bradycardia. All these patients were considered as failures and were referred to surgery. Femoral artery thrombosis occurred in 6 patients that improved after administration of unfractionated heparin. Aortic end was not fully covered in 4 patients, while stent protrusion to one of pulmonary arteries causing jailing of the other branch occurred in another two patients. Forty out of the 54 successful cases (74%) were discharged home while ICU mortality due to sepsis occurred in 11 patients and mortality related to the stenting procedure occurred in 3 patients (Table 1). Pulmonary atresia with intact ventricular septum was the most common cardiac anomaly (n = 24, 58.2%) followed by tricuspid atresia (n = 15).

**Group 2: PDA originating from the under-surface of aortic arch (n = 11).** Nine out of 11 patients underwent a successful procedure (81.8%). Eight out of the 9 patients (88.9%) were discharged home. The axillary artery was the most preferred access route (54.5%, n = 6) in this group. One patient developed acute stent thrombosis which required urgent MBT shunt (Table 1). Most of patients in this group had pulmonary atresia with VSD (n = 4), followed by heterotaxy syndrome (n = 3), and PDA was tortuous in 9 patients.

**Group 3: PDA originating opposite the left subclavian artery (n = 17).** Successful stenting was achieved in 14 out of 17 patients (82.35%). The PDA was tortuous in 13 patients (76.5%) and the most preferred access route was the axillary artery (64.7%, n = 11). 9/14 patients were discharged home (64%). One patient developed acute stent thrombosis and required urgent surgical shunt (Table 1). Most of patients within this group had pulmonary atresia with VSD (41%, n = 7) or a single ventricle (29.4%, n = 5).

**Group 4: PDA originating opposite the innominate artery/brachiocephalic artery (n = 1).** In this patient with pulmonary atresia and VSD, the PDA was accessed through an axillary approach. The procedure was successful without complications and the patient was discharged home 6 days after procedure (Table 1).

**Analysis of outcome according to procedure approach.** Success was achieved in 60/72 cases (83%) using the femoral approach compared to 13/19 cases (68%) using the axillary approach. All of the 3 cases treated from the carotid approach, and all of the 2 cases approached from the femoral vein were stented successfully.

**Indicators of PDA stenting success.** On comparing the successfully-treated versus the failed cases, it was noted that success was associated with straight PDA, larger ductal diameter, and younger age. The failed cases had higher exposure to radiation and prolonged hospital stays (Table 2). Logistic regression was performed for evaluation of indicators of successful procedure: patients with a straight ductus were 8 times more likely to be successful; likewise, younger patients and those with larger PDAs had higher odds of a successful procedure (Table 3).

**Follow-up of patients who underwent successful PDA stenting.** In our series, six (7.7%) patients in the successful group had minor displacement of the stent into one of the pulmonary artery branches (Fig 2H), three of them needed arterioplasty during cardiac surgery. Most of PDAs were stented using a single stent but sometimes more than one stent could be needed to cover the whole length of PDA. In our cohort, we had 6 patients whose aortic end of the PDA was not covered by the stent; 3 of them underwent successful stenting 3–6 days after the initial procedure and 2 of them maintained saturations >75% and discharged then underwent MBT shunt 2 months later. One patient had failed trials for insertion of another stent then developed desaturation and was referred to surgery. Fifty-eight patients (74%) survived to hospital discharge after a mean period of 4 days (3–9 days). The main cause of mortality was sepsis (90%). A total of 43 patients were followed-up for a period of 3–12 months: whereas 10 patients died in the interim; and 5 patients dropped follow-up. Oxygen saturation ranged from 79–90% showing gradual decline over time (Fig 2). MDCT angiography was performed whenever

**Table 2. Comparison between groups with successful and the failed procedures.**

|  | Successful group (n = 78) | Failed group (n = 18) | P- value |
|---|---|---|---|
| **Age, days** | 19.50(10.00–36.00) | 31.50(19.00–60.00) | 0.044* |
| **Weight, Kg** | 3.05(2.80–3.50) | 3.15(2.67–4.00) | 0.654 |
| **RPA z score** | -1.02(-2.13- -0.21) | -1.57(-2.36- -0.09) | 0.420 |
| **LPA z score** | -1.00(-1.91- -.25) | -1.31(-2.02- -0.39) | 0.611 |
| **PDA at aortic end, mm** | 4.80(4.00–5.00) | 4.50(3.35.5.50) | 0.603 |
| **PDA at pulmonary end, mm** | 3.00(2.5–3.50) | 2.50(2.00–3.00) | 0.021* |
| **PDA length, mm** | 15.80(14.00–18.50) | 16.30(15.00–20.00) | 0.327 |
| **PDA shape, n (%)** |  |  | 0.023* |
| Tortuous | 53.00(67.90) | 17.00(94.40) |  |
| Straight | 25.00(32.10) | 1.00(5.60) |  |
| **Stent diameter, mm** | 3.5(3.5–4) | 3.5(3–4) | 0.570 |
| **Stent length, mm** | 18(18–24) | 24(16–24) | 0.406 |
| **Fluoroscopy time (minutes)** | 17.18(10.60–29.37) | 29.95(25.99–43.25) | 0.0004* |
| **Radiation exposure (Gycm$^2$)** | 34.00(18.00–73.00) | 68.00(51.50–104.50) | 0.005* |
| **Complications** |  |  |  |
| Femoral artery thrombosis, n (%) | 5.00(6.40) | 2.00(11.10) |  |
| Stent embolization, n (%) |  | 2.00(11.10) |  |
| Stent thrombosis, n (%) |  | 2.00(11.10) |  |
| Displacement to pulmonary arteries, n (%) | 6.00(7.70) |  |  |
| Uncovered aortic end, n, % | 5.00(6.40) | 1.00(5.55) |  |
| **Hospital stay, days** | 4(3–10) | 15(6–31) | 0.011* |

RPA: right pulmonary artery, LPA left pulmonary artery, PDA: patent ductus arteriosus.

oxygen saturation falls beneath 75%, or prior to planned surgical intervention. In our cohort, this imaging modality was done for all of the 43 cases at a mean of 4 months post-procedure and showed significant growth of pulmonary arteries (Table 4). *Surgical palliation and repair*: Nine cases underwent partial cavo pulmonary connection (PCPC), 6 cases underwent Modified Blalock-Taussig Thomas shunt and 3 cases underwent biventricular repair (Fig 3).

**Table 3. Indicators of success for PDA stenting.**

| Predictors | OR | 95% C.I. | p value |
|---|---|---|---|
| **Straight PDA[1]** | 8.019 | 1.011–63.68 | 0.048* |
| **Weight** | 0.681 | 0.311–1.489 | 0.335 |
| **Age** | 0.978 | 0.960–0.998 | 0.031* |
| **Axillary access [2]** | 2.38 | 0.757–7.57 | 0.138 |
| **PDA from middle arch[3]** | 1.08 | 0.209–5.62 | 0.924 |
| **PDA opposite SCA [3]** | 1.12 | 0.281–4.49 | 0.869 |
| **PDA length** | 0.904 | 0.748–1.09 | 0.294 |
| **PDA size at Pulmonary end** | 2.94 | 1.172–7.402 | 0.022* |
| **Fluoroscopy time** | 0.959 | 0.925–0.994 | 0.021* |

[1]Compared to tortuous PDA

[2] compared to femoral axis

[3]compared to PDA from proximal descending aorta

*: statistically significant, O.R: odds ratio, CI: confidence interval, PDA: patent ductus arteriosus, SCA: subclavian artery.

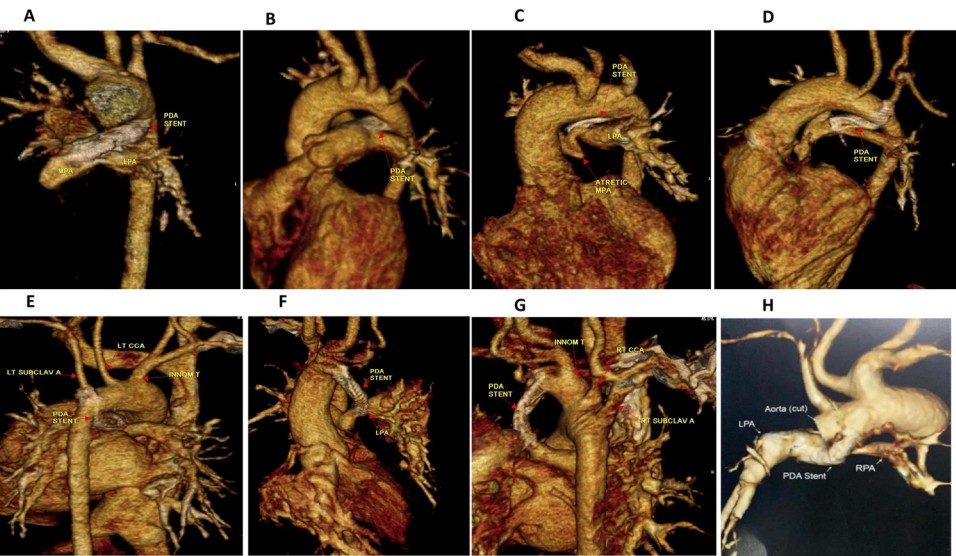

**Fig 2. Multidetector computed tomography with 3D volume rendering showing PDA stents in different PDA origins. A-B:** PDA from the proximal descending aorta, **C:** PDA from undersurface of aortic arch, **D-E**: PDA from opposite the origin of left subclavian artery, **F-G;**PDA arises opposite the innominate (Brachiocephalic trunk),**H:**PDA stent is seen protruded into the LPA causing jailing of RPA and associated with RPA origin stenosis. PDA: patent ductus arteriosus, MPA: Mean pulmonary artery, LPA: right pulmonary artery, LPA left pulmonary artery, CCA: common carotid artery.

## Discussion

A debate is ongoing on the preferred method of palliating cardiac anomalies with duct-dependent pulmonary circulation, which is turning in favor of ductal stenting. Better clinical outcomes including less procedural complications, less length of hospital stay, lower risk of diuretic use, better and more symmetrical pulmonary arterial growth [12, 13], and even economically [14] favoring PDA stent. Priority for ductal stenting will be in countries with limited resources [15], long waiting lists, with similar pulmonary arterial growth and distortion rates [12] but on the expense of a minor tendency for more frequent re-interventions. However, a surgical shunt is still necessary for certain patients [1] We sought to study the factors associated with procedural success, in the venue of different ductal morphologies and the utilization of alternate approaches than the classical femoral artery. We did not exclude the prematures less than 2.5 kg like other reports [8, 16–18] for the fear of vascular injury [19, 20]. In fact, we

**Table 4. Pulmonary arterial growth and oxygen saturation before and after PDA stenting.**

|  | Initially | Follow-up | P- value |
|---|---|---|---|
| **Pulmonary arteries growth based on MDCT angiography at age of 4 months** | | | |
|  | *Before Stenting* | *4 months after stenting* | |
| **RPA Z score** | -0.79(-1.91- -0.12) | 0.93(0.28–1.82) | <0.0001* |
| **LPA Z score** | -1.02(-1.82- -0.38) | 0.40(0.24–1.01) | <0.0001* |
| **ygen saturation before and immediately after PDA stenting** | | | |
|  | *Before PDA stenting* | *Immediately after PDA stenting* | |
| **Oxygen saturation** | 65.00(65.00–70.00) | 88.00(85.00–90.00) | <0.0001* |

RPA: right pulmonary artery, LPA left pulmonary artery, MDCT: Multidetector computed tomography, PDA: patent ductus arteriosus
*: Statistically significant.

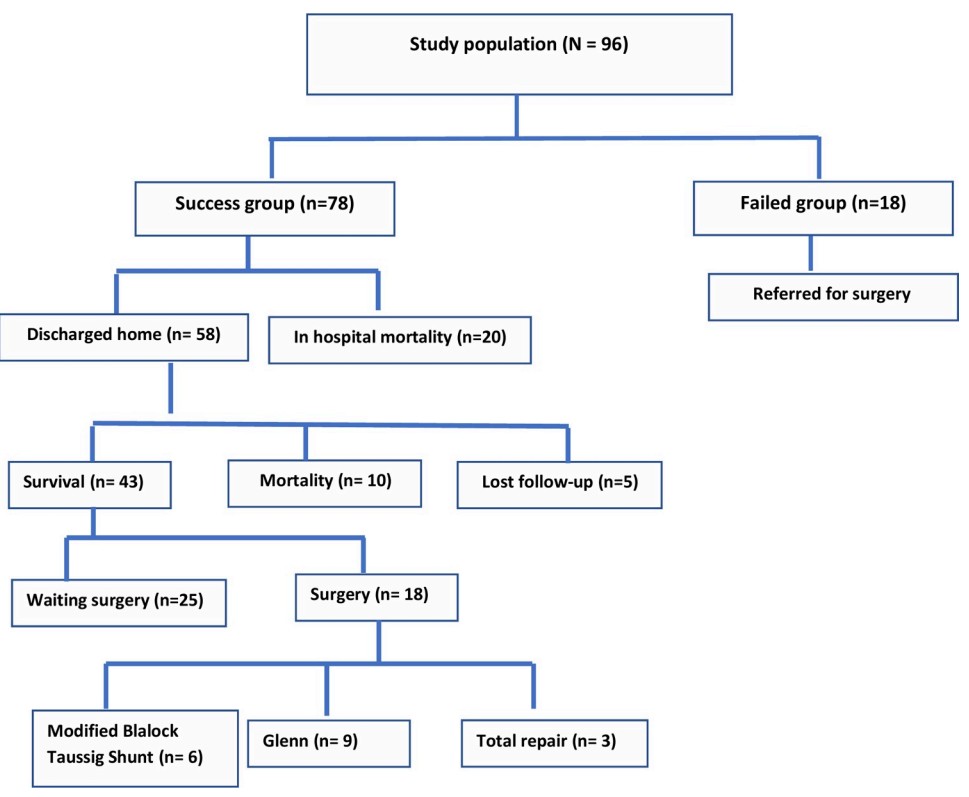

**Fig 3. Scheme of the studied population.**

did not have any major vascular complications in this group. Moreover, we report a higher success rate in association with younger age, which advocates for intervening as early as possible in these vulnerable neonates to minimize hospitalization and the related inherent risks of infection, ventilation and metabolic derangements that have a direct impact on the post-procedural outcome. The dose, not only the duration, of prostaglandin infusion might also be a risk factor [21]: proposed by the fact that larger ductuses (and therefore requiring less prostaglandin doses) were associated with a more favorable outcome. Pre-procedural assessment and planning is of utmost importance, the interventionist needs to define the ductal anatomy and select the most appropriate approach and prepare the necessary equipment [8, 15–18]. **PDA morphology**: A straight PDA was 8 times more likely to be successfully stented, compared to a tortuous PDA. The definition of PDA "tortuosity" is mainly subjective and most of the PDAs with higher degrees of tortuosity were seen arising from the underside of the arch [22]. Similar to several reports, most of our patients (72.9%) had a tortuous PDA [8, 15]. The overall success rate was 81.25%: which is comparable to previously reported success rates ranging from 80 to 100%, with the variation that some of these studies included either a small number of patients or excluded complex ductal morphologies [8, 15–18, 23]. Complex tortuosity especially multiple bends with acute angles is reported to propose higher failure rates of ductal stenting compared to others [23], but could be stented nevertheless [22]. In our study, most of patients who failed PDA stenting had tortuous PDAs compared to patients with successful PDA stenting (94.4% versus 67.9%, $p = 0.023$). **Vascular access**: The approach to stent the duct is a major determinant in the success of the procedure [22] as it allows direct access to cannulate the PDA and to obtain a stable wire position which in turn is reflected on the procedural time and radiation exposure. We advocate for femoral access to the PDAs arising from the proximal

descending Aorta; and the axillary approach to ductus from the undersurface of the arch and opposite to the left subclavian artery. Similarly, *Qureshi et al* reported significant association between ductal origin and vascular access choice [22]. **Radiation exposure:** In our series, fluoroscopy time and radiation exposure were lower in successful procedures compared to the failed ones. This could be attributed to the increased time allocated to PDA access attempts in failed cases. **Procedural complications**: Stent embolization and thrombosis is a major complication of ductal stenting procedures. In our cohort, 2 patients developed stent embolization and another 2 patients developed acute stent thrombosis. Some authors reported relatively higher incidence of stent embolization and acute thrombosis [17]. **PDA stent:** Length of the stent needs to be a few millimeters larger than the measured ductal length, to be able to cover both ends and avoid slipping during balloon inflation [19, 24], given that the inflated stent shortens by 15–20% [1, 25]. On the other hand, protrusion of the stent into one of the branch pulmonary arteries may lead to jailing of the other branch with subsequent stenosis or even disconnection from the main pulmonary artery, reported in up to 22% of cases and usually requires arterioplasty during surgical repair or palliation [22], which in our report was less frequent (7.7%). **Hospital stay**: In the current series, the in-hospital mortality was relatively higher compared to other reports (25.6%), that we attribute to the delayed referrals from across the country, and the unavailability of small-sized surgical shunts which forces proper patient selection to be overlooked.

## Follow-up after ductal stenting

In our cohort, initially, the oxygen saturation significantly increased immediately after PDA stenting then gradually decreased over time (median oxygen saturation at 6 months was 70%), with no significant difference compared to the post-procedural oxygen saturation, p 0.988). Whether due to neo-intimal proliferation [5, 26] or unpaired growth of the pulmonary vascular bed, as PDA stenting promotes the growth of pulmonary arteries and pulmonary vascular bed even with jailing of one of the pulmonary arteries. Data from cardiac catheterization done post-ductal-stenting showed balanced pulmonary arterial growth even with the presence of proximal stenosis at the insertion site of the PDA stent [18, 27, 28]. In our cohort, z-score of pulmonary arteries markedly increased 4 months post-ductal stenting based on MDCT data performed for 37 patients.

## Indicators of successful stenting

Patients with straight PDA were 8 times more likely to get successful PDA stenting compared to patients with tortuous PDA. Younger age patients and those who had a relatively larger PDA diameter at pulmonary end had higher odds for successful procedure. This could be explained by the fact that patients with prolonged hospital stay had more complicated hospital courses compared to patients with early stenting before development of complications. The patients with larger PDAs at pulmonary end may have allowed for better access of the PDA as they may ease the passages of wires and catheters. Finally, ductal stenting has a learning curve, with time and more practice significant improvement of outcome even in the complex cases could be achieved.

## Limitations

Selection of patients could not be clearly set as frequently PDA stenting is performed as a lifesaving procedure when no surgical shunts are available. Although this work is considered one of the largest series of ductal stenting via different accesses in a relatively short period, the numbers during follow-up are small due to high percentage of dropout cases, and longer

follow-up is needed. This is a single-centered study, although receiving referrals from all over the country, but lacked the diversity of different operating teams. We did not record the dose or duration of prostaglandin infusion prior to the procedures, which could be an attributing factor to the final outcome. We suggest further studies on the effect of prostaglandins on the outcome.

## Conclusion

The success of the PDA stenting depends mainly on three critical factors: the straight morphology of the ductus, younger age at the time of the procedure and a relatively larger PDA diameter at the pulmonary end. The access route for percutaneous PDA stenting is markedly determined by the ductal origin and morphology.

## Author Contributions

**Conceptualization:** Hala Mounir Agha.

**Data curation:** Ola Kamel, Amal El-Sisi.

**Formal analysis:** Hala Mounir Agha, Ola Kamel, Amira Esmat, Rodina Sobhy.

**Investigation:** Hala Mounir Agha, Sahar S. Sheta.

**Methodology:** Hala Mounir Agha, Osama Abd -El Aziz, Ola Kamel, Sahar S. Sheta, Amal El-Sisi, Aya Fatouh.

**Supervision:** Hala Mounir Agha, Osama Abd -El Aziz, Sonia El-Saiedi, Rodina Sobhy.

**Writing – original draft:** Gaser Abdelmohsen.

**Writing – review & editing:** Hala Mounir Agha, Baher Hanna, Mai Hussien.

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
