## [Decision Letter · Decision Letter 0]

26 Aug 2020

PONE-D-20-24720

PDA stenting in duct-dependent pulmonary circulation: Effects of ductal morphology and origin on procedure approach and outcome

PLOS ONE

Dear Dr. Agha,

Thank you for submitting your manuscript to PLOS ONE. After careful consideration, we feel that it has merit but does not fully meet PLOS ONE’s publication criteria as it currently stands. Therefore, we invite you to submit a revised version of the manuscript that addresses the points raised during the review process.

Please address the reviewers' questions.

We look forward to receiving your revised manuscript.

Kind regards,

Academic Editor

PLOS ONE

Journal Requirements:

2. Please amend your current ethics statement to address the following concerns: Please explain why written consent was not obtained, how you recorded/documented participant consent, and if the ethics committees/IRBs approved this consent procedure.

3. Please modify the title to ensure that it is meeting PLOS’ guidelines (https://journals.plos.org/plosone/s/submission-guidelines#loc-title). In particular, the title should be specific, descriptive, concise, and comprehensible to readers outside the field.

4.In your Data Availability statement, you have not specified where the minimal data set underlying the results described in your manuscript can be found. PLOS defines a study's minimal data set as the underlying data used to reach the conclusions drawn in the manuscript and any additional data required to replicate the reported study findings in their entirety. All PLOS journals require that the minimal data set be made fully available. For more information about our data policy, please see http://journals.plos.org/plosone/s/data-availability.

5.Please include your tables as part of your main manuscript and remove the individual files. Please note that supplementary tables (should remain/ be uploaded) as separate "supporting information" files.

Additional Editor Comments (if provided):

Please address the reviewers' questions.

Reviewers' comments:

Reviewer's Responses to Questions

**Comments to the Author**

1. Is the manuscript technically sound, and do the data support the conclusions?

Reviewer #1: Yes

Reviewer #2: Partly

2. Has the statistical analysis been performed appropriately and rigorously? 

Reviewer #1: Yes

Reviewer #2: Yes

3. Have the authors made all data underlying the findings in their manuscript fully available?

Reviewer #1: Yes

Reviewer #2: No

4. Is the manuscript presented in an intelligible fashion and written in standard English?

Reviewer #1: Yes

Reviewer #2: No

5. Review Comments to the Author

Reviewer #1: The topic is not novel, but the approach is original and of relevance. However there are some critical aspects that need to be clarified.

Latest evidences are not in favour of PDA stenting in univentricular heart. Thus inclusion criteria should be weel discussed. Is a policy of author's center perform so many PDA stenting. How the Center's volume?

Introduction and Discussion: discuss critically the advantages/disavantages of PDA stenting, according to current leterature

Results:

-data should be discussed firt all together and then divided into subgroups

-Also in the abstract a short summary of global success/complication rate is missing

-Is there a surgical group for comparison?

Limitations section: should be much more robust. Patient's selection criteria is a clear limitation. The choice of the material seems also to be quite arbitrary

Limitations and discussion

Results are in line with previous publication, but complication rate is high. Why to prefer PDA stent to m-BT shunt?

Reviewer #2: Agha et al. Report on a relatively large cohort of consecutive neonates undergone PDA stenting for duct-dependant pulmonary circulation congenital lesion. The concept and the main topic of this article (proper vascular access depeding upon duct morphology) is not new and it has been extensively reported.

Percutaneous axillary artery approach for ductal stenting in critical right ventricular outflow tract lesions in the neonatal period

Colm R Breatnach 1, Varun Aggarwal 2, Khalid Al-Alawi 1, Colin J McMahon 1, Orla Franklin 1, Terence Prendiville 1, Paul Oslizlok 1, Kevin Walsh 1, Athar M Qureshi 2, Damien Kenny 1

Classification scheme for ductal morphology in cyanotic patients with ductal dependent pulmonary blood flow and association with outcomes of patent ductus arteriosus stenting.

Qureshi AM, Goldstein BH, Glatz AC, Agrawal H, Aggarwal V, Ligon RA, McCracken C, McDonnell A, Buckey TM, Whiteside W, Metcalf CM, Petit CJ

Comparison Between Patent Ductus Arteriosus Stent and Modified Blalock-Taussig Shunt as Palliation for Infants With Ductal-Dependent Pulmonary Blood Flow: Insights From the Congenital Catheterization Research Collaborative.

Glatz AC, Petit CJ, Goldstein BH, Kelleman MS, McCracken CE, McDonnell A, Buckey T, Mascio CE, Shashidharan S, Ligon RA, Ao J, Whiteside W, Wallen WJ, Metcalf CM, Aggarwal V, Agrawal H, Qureshi AM.

Circulation. 2018 Feb 6;137(6):589-601. doi: 10.1161/CIRCULATIONAHA.117.029987. Epub 2017 Oct 17.

PMID: 29042354

In addition this reviewer has significant issues regarding this paper:

- Detailed information regarding the sub-group of patients with failed PDA stenting (18 patients, 19%) should be provided, such as specific reason for failure

- There is a concerning high rate of complication (including vascular and stent displacement), this should be discussed and explained

- There is no specific protocol described regarding the institutional approach to duct-dependent lesion. More specifically, is every patient with duct-dependant circulation referred to your insititution considered for PDA stent, irrespective of other parameters? Do you select patient referred for transcatether intervention?

- Do you have a surgical comparins group undergone BT shunt during the same period. Longterm comparison of such groups would be of some interest

- Do you have any follow-up surgical data at the time of corrective surgery regarding this patient population?

- Did you ever change vascular access after baseline angiography? In other words, were you always able to correctly identify PDA morphology and establish the correct vascular approach using pre-procedural echocardiography?

- There are significant procedural data missing: type, timing and dosing of anticoagulation, description of angiography, type of wire used to deploy stent.

- Long term follow up data are essential

6. PLOS authors have the option to publish the peer review history of their article (what does this mean?). If published, this will include your full peer review and any attached files.

Reviewer #1: **Yes: **Massimiliano Cantinotti

Reviewer #2: No

---

## [Author Response · Author response to Decision Letter 0]

8 Jan 2021

Rebuttal Letter

The authors thank the editors and the reviewers for their valuable comments, this letter includes a response to the individual remarks

General Comments

1. The title was adjusted to “Margin between success and failure of PDA stenting for duct-dependent pulmonary circulation” according to the journal’s requirements

2. Data availability and minimal data set: The main Excel is added as supplement

3. Tables included in the main manuscript

4. Figure legends added at end of manuscript

5. Authorship: the author (Baher Hanna) and (Mai Hussein) that were missed in the initial manuscript were re-added. And their contribution to the manuscript was added.

6. Definition of a successful procedure was added under “patients and methods”

Reviewer One:

- Latest evidences are not in favor of PDA stenting in univentricular heart. Thus inclusion criteria should be well discussed. Is a policy of author's center perform so many PDA stenting. How the Center's volume?

We added 3 extra references (25-28) that compare PDA stents to surgical shunts that tend to favor PDA stenting. Our institution performs around 900 catheterization cases per year: on average 50 of them are PDA stenting. Over the past 3 years, the small sized 3 and 3.5mm surgical shunts have been very sparse in the country and we resolve to PDA stenting as a life-saving procedure even in some cases in which a BT shunt would have been the choice if available.

- Introduction and Discussion: discuss critically the advantages/disadvantages of PDA stenting, according to current literature

The first paragraph under “Discussion” was edited as such

- Also in the abstract a short summary of global success/complication rate is missing

Added to the abstract

- Results: data should be discussed first all together and then divided into subgroups

The section “Results” was edited accordingly

- Is there a surgical group for comparison?

No, the aim of our study was to evaluate the different approaches for PDA stenting and the parameters reflecting the outcome in this specific cohort, rather than comparing results to surgical shunts. As mentioned earlier, in our country we are commonly obliged to perform a life-saving intervention due to the unavailability of small sized BT shunts.

- Limitations and discussion: Results are in line with previous publication, but complication rate is high. Why to prefer PDA stent to m-BT shunt?

In our institution, complications of neonatal MBTS are higher than PDA stents. A study to compare both is yet to be done.

- Limitations section: should be much more robust. Patient's selection criteria is a clear limitation. The choice of the material seems also to be quite arbitrary

The “Limitations” section was edited accordingly

Reviewer Two:

- The concept and the main topic of this article (proper vascular access depending upon duct morphology) is not new and it has been extensively reported.

We are mainly concerned with the indicators of a successful procedure based on the ductal morphology, origin and approach

- Detailed information regarding the sub-group of patients with failed PDA stenting (18 patients, 19%) should be provided, such as specific reason for failure

Results have been added to contain detailed information on the subgroups, details for procedural failure are discussed under “results” and Supplement Tables 

- There is a concerning high rate of complication (including vascular and stent displacement), this should be discussed and explained

This was mentioned in “results” The main cause of procedural failure was inadequate parked coronary wire inside one of the branch of pulmonary arteries (14 cases; 77.7%), while 2 cases (11.1%) were complicated by acute stent thrombosis, and another 2 cases with stent dislodgment. Other procedural complications comprised femoral artery thrombosis in 7 cases (7.2%), No patients died on table or within 6 hours of the procedure.

- There is no specific protocol described regarding the institutional approach to duct-dependent lesion. More specifically, is every patient with duct-dependant circulation referred to your institution considered for PDA stent, irrespective of other parameters? Do you select patient referred for transcatether intervention?

In our institution, we are frequently obliged to perform life-saving PDA stenting because of the unavailability of small sized surgical shunts in the country. This was mentioned under “limitations”.

- Do you have a surgical comparisons group undergone BT shunt during the same period. Long-term comparison of such groups would be of some interest

No we currently do not, such a study is yet to be done.

- Do you have any follow-up surgical data at the time of corrective surgery regarding this patient population?

Fifty-eight patients (74%) survived to hospital discharge after a mean period of 4 days (3-9 days). A total of 43 patients were followed-up for a period of 3-12 months: whereas 10 patients died; and 5 patients dropped follow-up. Oxygen saturation ranged from 79-90% showing gradual decline over time (figure 2). MDCT angiography was performed whenever oxygen saturation falls beneath 75%, or prior to planned surgical intervention. In our cohort, this imaging modality was done at a mean of 4 months post-procedure and showed significant growth of pulmonary arteries (Table 3). Surgical palliation and repair: 9 cases underwent partial cavo pulmonary connection (PCPC), 6 cases underwent Modified Blalock Taussig Thomas shunt and 3 cases underwent biventricular repair (Figure 3).

- Did you ever change vascular access after baseline angiography? In other words, were you always able to correctly identify PDA morphology and establish the correct vascular approach using pre-procedural echocardiography?

It is our routine to perform an initial aortogram through the femoral artery, accordingly to which the approach is decided. In 2 cases the approach was switched during the procedure. This is mentioned under “results”.

Based on ductal origin and morphology, the approach to ductal stenting was femoral artery in 72 patients (75%); axillary artery in 19 patients (19.8%), carotid artery in 3 patients (3.1%), and femoral vein in 2 patients (2.1%). It is of note that 2 cases that were initiated for femoral artery approach then shifted to the axillary route

- There are significant procedural data missing: type, timing and dosing of anticoagulation, description of angiography, type of wire used to deploy stent.

These data were added under “patients and methods”

Procedures: The femoral arterial access was obtained in all patients by 4 French short sheath, 100 units/kg unfractionated intravenous heparin and prophylactic antibiotic was given to every patient followed by aortogram using 4 French pigtail catheter. Patients were divided into 4 groups according to PDA origin: Group 1 with PDA from proximal descending aorta, Group 2 with PDA from undersurface of aortic arch, Group 3 with PDA arising opposite the subclavian artery and Group 4 with PDA arising opposite the innominate/brachiocephalic artery [figure1A-H]. Based on this angiogram data, the length of the stent and its diameter was selected as well as the best access route, i.e. whether to proceed with the femoral arterial access or to convert to another approach like axillary, carotid or femoral vein access. Two different types of catheters either 4 French cut pigtail or 4 French right Judkin [JR]were used for encroaching of PDA. PDA was crossed with 0.014 inch percutaneous transluminal coronary angioplasty [PTCA] wire [Middle Support Weight guide wire, Boston Scientific, USA] and it was parked in either of the branch pulmonary arteries . In all cases, pre-mounted bare metal stents [architect, Life Vascular Device Biotech, Barcelona, Spain] or drug-eluting coronary stents [Monorail, Everolimus-Eluting Platinum Chromium, Boston Scientific, USA]. The diameter was chosen with respect to the weight of the patient, size of the PDA, its constricted segment and branch pulmonary arteries. Length of stent was chosen to cover the entire length of the duct with proximal end into aorta and distal end into branch PA..The selected coronary stent was tracked over PTCA wire and placed across the PDA with its proximal end in the aorta and distal end in branch pulmonary artery. After confirming the position of the stent, it was deployed with its initial nominal pressure then re-dilated for better apposition. Balloon was removed after deployment keeping the wire in situ. Injection was done again to check the position of the stent, flow across the stent and branch pulmonary arteries. After achieving satisfactory results, wire was removed. In cases of branch pulmonary artery origin stenosis, we aimed to deploy the stent distal to the stenosed segment. After stent implantation, angiogram injection was performed to confirm the stent position. All the patients were shifted to ICU and received unfractionated heparin infusion on 20 IU/kg/hr for 24-48 hours (partial prothrombin time was maintained 2–3 times of normal). All the patients were given antiplatelets in form of aspirin on 5mg/kg/day that was continued until undergoing second-stage palliation or repair. Procedural success of PDA stenting was defined as the implantation of a well-seated stent, good flow for both pulmonary arteries and stable SpO2 above 75% for a hemodynamic-stable patient.

- Long term follow up data are essential

Our study’s end-point was the 2nd surgical intervention; either the cavo-pulmonary shunt for univentricular heart or biventricular repair as definitive surgery.

Regarding Data Availability statement, I have specified where the minimal data set underlying the results described in the manuscript can be found in . 

https://doi.org/10.5061/dryad.ht76hdrdk.

Margin between success and failure of PDA stenting for duct-dependent pulmonary circulation submitted with https://doi.org/10.5061/dryad.ht76hdrdk. There may be a delay for processing before the item is available.

---

## [Decision Letter · Decision Letter 1]

21 Jan 2021

PONE-D-20-24720R1

Margin between success and failure of PDA stenting for  duct-dependent pulmonary circulation

PLOS ONE

Dear Dr. Agha,

Thank you for submitting your manuscript to PLOS ONE. After careful consideration, we feel that it has merit but does not fully meet PLOS ONE’s publication criteria as it currently stands. Therefore, we invite you to submit a revised version of the manuscript that addresses the points raised during the review process.

Please revise accordingly. 

We look forward to receiving your revised manuscript.

Kind regards,

Academic Editor

PLOS ONE

Reviewers' comments:

Reviewer's Responses to Questions

**Comments to the Author**

1. If the authors have adequately addressed your comments raised in a previous round of review and you feel that this manuscript is now acceptable for publication, you may indicate that here to bypass the “Comments to the Author” section, enter your conflict of interest statement in the “Confidential to Editor” section, and submit your "Accept" recommendation.

Reviewer #3: (No Response)

Reviewer #4: All comments have been addressed

2. Is the manuscript technically sound, and do the data support the conclusions?

Reviewer #3: Yes

Reviewer #4: Yes

3. Has the statistical analysis been performed appropriately and rigorously? 

Reviewer #3: I Don't Know

Reviewer #4: Yes

4. Have the authors made all data underlying the findings in their manuscript fully available?

Reviewer #3: Yes

Reviewer #4: No

5. Is the manuscript presented in an intelligible fashion and written in standard English?

Reviewer #3: Yes

Reviewer #4: Yes

6. Review Comments to the Author

Reviewer #3: This report details a remarkably large series of ductal – dependent pulmonary circulation cases palliated with PDA stenting from a Low-Middle Income Country. Over 2 years, they collected 96 consecutive cases with an 81% early success rate, for which they should be congratulated. However, due to a lack of simultaneous surgical shunt availability, they were unable to compare this series to anything except external historical controls. Thus, this is intrinsically just a description of a cohort from a single center having a single procedure, and not a scientific comparison study. Although the procedures reported are not novel or new, yet the group was able to discern predictors (risk factors) for success or failure such that better outcomes occurred in younger patients with straighter and larger ducts. Given their resource limitations, it is perhaps not surprising that only 74% of cases survived to hospital discharge, and that 90% of deaths were due to sepsis.

Several comments can be made. Firstly, regarding statistics, it is noted that of the 4 groups of anatomical types, group 4 had only 1 patient. It is not certain that in this case of pulmonary atresia/VSD with a “duct” from the innominate artery, wasn’t really a MAPCA being stented. In any case, the statistics in Table 1 give p values after 4 groups. How was the n=1 group handled? Please explain and consider deleting group 4 as a single exception, or adding the case to group 3. In fact, the 4 anatomical groups are a little unconventional and need a reference to the literature (unless I missed that).

Please state if there was any difference between bare metal and everolimus drug eluting stent groups.

The smallest size patient appears to be 2.1 kg in the Table, and the vascular complication rate is not excessive in light of 12 patients under 2.5 kg. Can the authors expound in the Discussion on their recommendations for smallest body size acceptable, and if their anticoagulation protocol changes with different risk factors.

There is no long - term follow up beyond 1 year, as noted in the study limitations.

The Discussion is long for a single case series and the authors should try to be more terse. They should emphasis the uniqueness of this series:

1. This was accomplished in a country with limited resources in regard to surgical shunts.

2. The use of a short 4F femoral artery sheath was associated with fewer than expected vascular complications.

3. The selection of the axillary/other approach after careful analysis was associated with successful outcomes.

Please discuss the differentiation of tortuous duct coming from an unusual site of origin versus MAPCA in the presence of pulmonary atresia.

The authors appear to give some results for the first time in the Discussion section (branch PA stent protrusion, oxygen saturation trends), which should be stated clearly in the results section first.

The Limitations section is good.

The Track Changes I received in the first version of the manuscript were not complete, but were, however, well explained in the Rebuttal letter.

The paper is well written but does need some editing of English language. The Figures and Tables are good and Supplemental materials complete. The References are relevant.

Specific Comments:

1. References #25 and #28 appears to be identical and must be corrected.

2. Line 237 on page 9 – Start a new paragraph here.

3. Where are the Figures 2 and 3 referred to on page 10?

Reviewer #4: Dear Authors

We sit every week in conference and debate this question

You have a large cohort but gave us very little data , if I need to decide for a patient what rout to go - BT shunt or stunting, please help me

here are some issues you need to address

1. Why have you decided to go for a stent, that is from how many total patients did you decide to go for a BT shunt and how many were referred for stenting? What were the criteria for each group?

2. What were the comorbidities for each patient?

3. Did the comorbidities play a role in deciding who should go to each of the procedures?

4. What was the outcome between the BT shunt group and the stented group – and why?

Moreover, when looking at the data

5. What additional precedures were done (in the table there is a short list -pulm valvuloplasty, septectomy and more) – did it change the procedure, were the outcomes different, do you recommend these elaborate long procedures?

6. What is tortuous? How long, what was your definition? Rarely do we see a completely straight PDA, most of the have bents and more – how did you address this, what and with what modality

7. What is failure – do you mean procedure aborted? When did you decide to abort 20% of the stenting- after the first fluoroscopy? Ofter you saw you cant pass through? Any other special complications?

8. What was your mode of device selection and when did you decide to put 2 stents in one PDA?

Make this into the Cairo Guidelines for smaller programs which may use your knowledge

7. PLOS authors have the option to publish the peer review history of their article (what does this mean?). If published, this will include your full peer review and any attached files.

Reviewer #3: No

Reviewer #4: **Yes: **Shai Tejman-Yarden MD Mac MBA

---

## [Author Response · Author response to Decision Letter 1]

21 Jun 2021

General Comments

1. The title was adjusted to “Margin between success and failure of PDA stenting for duct-dependent pulmonary circulation” according to the journal’s requirements

2. Data availability and minimal data set: The main Excel is added as supplement

3. Tables included in the main manuscript

4. Figure legends added at end of manuscript

5. Authorship: the author (Baher Hanna) and (Mai Hussein) that were missed in the initial manuscript were re-added. And their contribution to the manuscript was added.

6. Definition of a successful procedure was added under “patients and methods”

Reviewer One:

- Latest evidences are not in favor of PDA stenting in univentricular heart. Thus inclusion criteria should be well discussed. Is a policy of author's center perform so many PDA stenting. How the Center's volume?

We added 3 extra references (25-28) that compare PDA stents to surgical shunts that tend to favor PDA stenting. Our institution performs around 900 catheterization cases per year: on average 50 of them are PDA stenting. Over the past 3 years, the small sized 3 and 3.5mm surgical shunts have been very sparse in the country and we resolve to PDA stenting as a life-saving procedure even in some cases in which a BT shunt would have been the choice if available.

- Introduction and Discussion: discuss critically the advantages/disadvantages of PDA stenting, according to current literature

The first paragraph under “Discussion” was edited as such

- Also in the abstract a short summary of global success/complication rate is missing

Added to the abstract

- Results: data should be discussed first all together and then divided into subgroups

The section “Results” was edited accordingly

- Is there a surgical group for comparison?

No, the aim of our study was to evaluate the different approaches for PDA stenting and the parameters reflecting the outcome in this specific cohort, rather than comparing results to surgical shunts. As mentioned earlier, in our country we are commonly obliged to perform a life-saving intervention due to the unavailability of small sized BT shunts.

- Limitations and discussion: Results are in line with previous publication, but complication rate is high. Why to prefer PDA stent to m-BT shunt?

In our institution, complications of neonatal MBTS are higher than PDA stents. A study to compare both is yet to be done.

- Limitations section: should be much more robust. Patient's selection criteria is a clear limitation. The choice of the material seems also to be quite arbitrary

The “Limitations” section was edited accordingly

Reviewer Two:

- The concept and the main topic of this article (proper vascular access depending upon duct morphology) is not new and it has been extensively reported.

We are mainly concerned with the indicators of a successful procedure based on the ductal morphology, origin and approach

- Detailed information regarding the sub-group of patients with failed PDA stenting (18 patients, 19%) should be provided, such as specific reason for failure

Results have been added to contain detailed information on the subgroups, details for procedural failure are discussed under “results” and Supplement Tables 

- There is a concerning high rate of complication (including vascular and stent displacement), this should be discussed and explained

This was mentioned in “results” The main cause of procedural failure was inadequate parked coronary wire inside one of the branch of pulmonary arteries (14 cases; 77.7%), while 2 cases (11.1%) were complicated by acute stent thrombosis, and another 2 cases with stent dislodgment. Other procedural complications comprised femoral artery thrombosis in 7 cases (7.2%), No patients died on table or within 6 hours of the procedure.

- There is no specific protocol described regarding the institutional approach to duct-dependent lesion. More specifically, is every patient with duct-dependant circulation referred to your institution considered for PDA stent, irrespective of other parameters? Do you select patient referred for transcatether intervention?

In our institution, we are frequently obliged to perform life-saving PDA stenting because of the unavailability of small sized surgical shunts in the country. This was mentioned under “limitations”.

- Do you have a surgical comparisons group undergone BT shunt during the same period. Long-term comparison of such groups would be of some interest

No we currently do not, such a study is yet to be done.

- Do you have any follow-up surgical data at the time of corrective surgery regarding this patient population?

Fifty-eight patients (74%) survived to hospital discharge after a mean period of 4 days (3-9 days). A total of 43 patients were followed-up for a period of 3-12 months: whereas 10 patients died; and 5 patients dropped follow-up. Oxygen saturation ranged from 79-90% showing gradual decline over time (figure 2). MDCT angiography was performed whenever oxygen saturation falls beneath 75%, or prior to planned surgical intervention. In our cohort, this imaging modality was done at a mean of 4 months post-procedure and showed significant growth of pulmonary arteries (Table 3). Surgical palliation and repair: 9 cases underwent partial cavo pulmonary connection (PCPC), 6 cases underwent Modified Blalock Taussig Thomas shunt and 3 cases underwent biventricular repair (Figure 3).

- Did you ever change vascular access after baseline angiography? In other words, were you always able to correctly identify PDA morphology and establish the correct vascular approach using pre-procedural echocardiography?

It is our routine to perform an initial aortogram through the femoral artery, accordingly to which the approach is decided. In 2 cases the approach was switched during the procedure. This is mentioned under “results”.

Based on ductal origin and morphology, the approach to ductal stenting was femoral artery in 72 patients (75%); axillary artery in 19 patients (19.8%), carotid artery in 3 patients (3.1%), and femoral vein in 2 patients (2.1%). It is of note that 2 cases that were initiated for femoral artery approach then shifted to the axillary route

- There are significant procedural data missing: type, timing and dosing of anticoagulation, description of angiography, type of wire used to deploy stent.

These data were added under “patients and methods”

Procedures: The femoral arterial access was obtained in all patients by 4 French short sheath, 100 units/kg unfractionated intravenous heparin and prophylactic antibiotic was given to every patient followed by aortogram using 4 French pigtail catheter. Patients were divided into 4 groups according to PDA origin: Group 1 with PDA from proximal descending aorta, Group 2 with PDA from undersurface of aortic arch, Group 3 with PDA arising opposite the subclavian artery and Group 4 with PDA arising opposite the innominate/brachiocephalic artery [figure1A-H]. Based on this angiogram data, the length of the stent and its diameter was selected as well as the best access route, i.e. whether to proceed with the femoral arterial access or to convert to another approach like axillary, carotid or femoral vein access. Two different types of catheters either 4 French cut pigtail or 4 French right Judkin [JR]were used for encroaching of PDA. PDA was crossed with 0.014 inch percutaneous transluminal coronary angioplasty [PTCA] wire [Middle Support Weight guide wire, Boston Scientific, USA] and it was parked in either of the branch pulmonary arteries . In all cases, pre-mounted bare metal stents [architect, Life Vascular Device Biotech, Barcelona, Spain] or drug-eluting coronary stents [Monorail, Everolimus-Eluting Platinum Chromium, Boston Scientific, USA]. The diameter was chosen with respect to the weight of the patient, size of the PDA, its constricted segment and branch pulmonary arteries. Length of stent was chosen to cover the entire length of the duct with proximal end into aorta and distal end into branch PA..The selected coronary stent was tracked over PTCA wire and placed across the PDA with its proximal end in the aorta and distal end in branch pulmonary artery. After confirming the position of the stent, it was deployed with its initial nominal pressure then re-dilated for better apposition. Balloon was removed after deployment keeping the wire in situ. Injection was done again to check the position of the stent, flow across the stent and branch pulmonary arteries. After achieving satisfactory results, wire was removed. In cases of branch pulmonary artery origin stenosis, we aimed to deploy the stent distal to the stenosed segment. After stent implantation, angiogram injection was performed to confirm the stent position. All the patients were shifted to ICU and received unfractionated heparin infusion on 20 IU/kg/hr for 24-48 hours (partial prothrombin time was maintained 2–3 times of normal). All the patients were given antiplatelets in form of aspirin on 5mg/kg/day that was continued until undergoing second-stage palliation or repair. Procedural success of PDA stenting was defined as the implantation of a well-seated stent, good flow for both pulmonary arteries and stable SpO2 above 75% for a hemodynamic-stable patient.

- Long term follow up data are essential

Our study’s end-point was the 2nd surgical intervention; either the cavo-pulmonary shunt for univentricular heart or biventricular repair as definitive surgery.

---

## [Decision Letter · Decision Letter 2]

7 Jul 2021

PONE-D-20-24720R2

Margin between success and failure of PDA stenting for  duct-dependent pulmonary circulation

PLOS ONE

Dear Dr. Agha,

Thank you for submitting your manuscript to PLOS ONE. After careful consideration, we feel that it has merit but does not fully meet PLOS ONE’s publication criteria as it currently stands. Therefore, we invite you to submit a revised version of the manuscript that addresses the points raised during the review process.

Please revise accordingly.

We look forward to receiving your revised manuscript.

Kind regards,

Academic Editor

PLOS ONE

Journal Requirements:

Reviewers' comments:

Reviewer's Responses to Questions

**Comments to the Author**

1. If the authors have adequately addressed your comments raised in a previous round of review and you feel that this manuscript is now acceptable for publication, you may indicate that here to bypass the “Comments to the Author” section, enter your conflict of interest statement in the “Confidential to Editor” section, and submit your "Accept" recommendation.

Reviewer #3: (No Response)

Reviewer #4: All comments have been addressed

Reviewer #5: (No Response)

2. Is the manuscript technically sound, and do the data support the conclusions?

Reviewer #3: Yes

Reviewer #4: Yes

Reviewer #5: Yes

3. Has the statistical analysis been performed appropriately and rigorously? 

Reviewer #3: I Don't Know

Reviewer #4: Yes

Reviewer #5: Yes

4. Have the authors made all data underlying the findings in their manuscript fully available?

Reviewer #3: Yes

Reviewer #4: Yes

Reviewer #5: Yes

5. Is the manuscript presented in an intelligible fashion and written in standard English?

Reviewer #3: Yes

Reviewer #4: Yes

Reviewer #5: No

6. Review Comments to the Author

Reviewer #3: My previous comments on this manuscript are not addressed in the response by the authors, so I have no further comments at this time.

Reviewer #4: The authors answered my questions and the paper has a huge impact on the field. I strongly advise to publish

Reviewer #5: I am not an expert at the first-version review, so I only comment on whether the author's answer to the question from the expert at the first-version review is adequate.

1) The purpose of the research: The abstract states that it is necessary to analyze the factors of the success of the operation. In fact, it is grouped and compared according to the origin of the PDA. In the following discussion, it is said that this study mainly wants to analyze the situation of different approaches and influence the success of the operation. The indicators that the author ultimately wants to be clear need to be consistent. If the main research in the discussion is the influence of different approaches on the success rate，why not group them using approaches?

2) In the method part, the second reviewer asked the author to give a detailed description of the inclusion and exclusion criteria. The author’s answer was not very detailed. Although there are intermittent descriptions in the method part of the article, such as weight, coagulation dysfunction, etc., it should be systematically Write down the inclusion and exclusion criteria for patients who actually undergo PDA stent placement; this is also a question raised by the reviewer in the limitation section.

3) The results part: there are overlaps description in results. The first two paragraphs are for the explanation of the patient's baseline data. There is no need to write approach groupings. Is it better to describe in the procedure part? And data in the table does not need to be described all them again. It is possible to group by PDA origin, but in fact, according to the author's results, the procedure approach, PDA distortion, and complications of groups 2-4 are similar, and can be used as a unified group for comparison with the first group. Regarding the follow-up part, 43 patients were followed up for 3-12 months. How many cases were actually followed up for 4 months? What is the actual number of people undergoing MDCT angiography 4 months after surgery? This is not described in detail. “The detailed information regarding the sub-group of patients with failed PDA stenting should be provided ”which was proposed by the second reviewer, was not discussed adequately.

7. PLOS authors have the option to publish the peer review history of their article (what does this mean?). If published, this will include your full peer review and any attached files.

Reviewer #3: No

Reviewer #4: **Yes: **Shai Tejman Yarden MD MSc MBA

Reviewer #5: No

---

## [Author Response · Author response to Decision Letter 2]

28 Sep 2021

1. If the authors have adequately addressed your comments raised in a previous round of review and you feel that this manuscript is now acceptable for publication, you may indicate that here to bypass the “Comments to the Author” section, enter your conflict of interest statement in the “Confidential to Editor” section, and submit your "Accept" recommendation.

Reviewer #3: (No Response)

Reviewer #4: All comments have been addressed

Reviewer #5: (No Response)

2. Is the manuscript technically sound, and do the data support the conclusions?

Reviewer #3: Yes

Reviewer #4: Yes

Reviewer #5: Yes

3. Has the statistical analysis been performed appropriately and rigorously?

Reviewer #3: I Don't Know

Reviewer #4: Yes

Reviewer #5: Yes

4. Have the authors made all data underlying the findings in their manuscript fully available?

Reviewer #3: Yes

Reviewer #4: Yes

Reviewer #5: Yes

5. Is the manuscript presented in an intelligible fashion and written in standard English?

Reviewer #3: Yes

Reviewer #4: Yes

Reviewer #5: No

The manuscript has been revised and the language edited

6. Review Comments to the Author

Reviewer #3: My previous comments on this manuscript are not addressed in the response by the authors, so I have no further comments at this time.

Reviewer #4: The authors answered my questions and the paper has a huge impact on the field. I strongly advise to publish

Reviewer #5: I am not an expert at the first-version review, so I only comment on whether the author's answer to the question from the expert at the first-version review is adequate.

1) The purpose of the research: The abstract states that it is necessary to analyze the factors of the success of the operation. In fact, it is grouped and compared according to the origin of the PDA. In the following discussion, it is said that this study mainly wants to analyze the situation of different approaches and influence the success of the operation. The indicators that the author ultimately wants to be clear need to be consistent. If the main research in the discussion is the influence of different approaches on the success rate，why not group them using approaches?

The approach to PDA stenting is directly related to its origin, this has been clarified in the final edition

2) In the method part, the second reviewer asked the author to give a detailed description of the inclusion and exclusion criteria. The author’s answer was not very detailed. Although there are intermittent descriptions in the method part of the article, such as weight, coagulation dysfunction, etc., it should be systematically Write down the inclusion and exclusion criteria for patients who actually undergo PDA stent placement; this is also a question raised by the reviewer in the limitation section.

As indicated in the “material and methods” section (lines 1-3), all cases presenting for ductal stenting during the 2 year study period were included, only excluding those with untreated sepsis or coagulopathy.

3) The results part: there are overlaps description in results. The first two paragraphs are for the explanation of the patient's baseline data. There is no need to write approach groupings. Is it better to describe in the procedure part? 

Done

And data in the table does not need to be described all them again. It is possible to group by PDA origin, but in fact, according to the author's results, the procedure approach, PDA distortion, and complications of groups 2-4 are similar, and can be used as a unified group for comparison with the first group.

The authors believe that grouping the PDA according to its origin would be more informative to the interventionist who is planning for the procedure, therefore needs to decode on the best approach based on the PDA anatomy.

Regarding the follow-up part, 43 patients were followed up for 3-12 months. How many cases were actually followed up for 4 months? What is the actual number of people undergoing MDCT angiography 4 months after surgery? This is not described in detail. “The detailed information regarding the sub-group of patients with failed PDA stenting should be provided ”which was proposed by the second reviewer, was not discussed adequately.

All cases were assessed routinely by MDCT angiography prior to Glenn, but not after the operation. This was adjusted in the results.

7. PLOS authors have the option to publish the peer review history of their article (what does this mean?). If published, this will include your full peer review and any attached files.

Do you want your identity to be public for this peer review? For information about this choice, including consent withdrawal, please see our Privacy Policy.

Reviewer #3: No

Reviewer #4: Yes: Shai Tejman Yarden MD MSc MBA

Reviewer #5: No

---

## [Decision Letter · Decision Letter 3]

11 Oct 2021

PONE-D-20-24720R3Margin between success and failure of PDA stenting for  duct-dependent pulmonary circulationPLOS ONE

Dear Dr. Agha,

Thank you for submitting your manuscript to PLOS ONE. After careful consideration, we feel that it has merit but does not fully meet PLOS ONE’s publication criteria as it currently stands. Therefore, we invite you to submit a revised version of the manuscript that addresses the points raised during the review process.

Please correct the errors if references uncovered previously.

We look forward to receiving your revised manuscript.

Kind regards,

Academic Editor

PLOS ONE

Journal Requirements:

Reviewers' comments:

Reviewer's Responses to Questions

**Comments to the Author**

1. If the authors have adequately addressed your comments raised in a previous round of review and you feel that this manuscript is now acceptable for publication, you may indicate that here to bypass the “Comments to the Author” section, enter your conflict of interest statement in the “Confidential to Editor” section, and submit your "Accept" recommendation.

Reviewer #3: (No Response)

Reviewer #5: All comments have been addressed

2. Is the manuscript technically sound, and do the data support the conclusions?

Reviewer #3: Yes

Reviewer #5: Yes

3. Has the statistical analysis been performed appropriately and rigorously? 

Reviewer #3: No

Reviewer #5: Yes

4. Have the authors made all data underlying the findings in their manuscript fully available?

Reviewer #3: Yes

Reviewer #5: Yes

5. Is the manuscript presented in an intelligible fashion and written in standard English?

Reviewer #3: Yes

Reviewer #5: Yes

6. Review Comments to the Author

Reviewer #3: This is the second revision I have reviewed after I sent comments for revision. Incredibly, the authors have still not addressed my suggestions. For example, References #25 and 28 are still identical. The fact that "Group 4" only contains 1 member and yet is dealt with statistically, is not acceptable.

I have nothing further to say, and will not render any other decisions. I WILL NOT REVIEW THIS MANUSCRIPT AGAIN.

Reviewer #5: All questions mentioned in previous review have received more detailed and serious responses.The writing of the article has also been thoroughly improved.

7. PLOS authors have the option to publish the peer review history of their article (what does this mean?). If published, this will include your full peer review and any attached files.

Reviewer #3: No

Reviewer #5: No

---

## [Author Response · Author response to Decision Letter 3]

19 Oct 2021

Rebuttal Letter

The authors thank the editors and the reviewers for their valuable comments, this letter includes a response to the individual remarks

1. If the authors have adequately addressed your comments raised in a previous round of review and you feel that this manuscript is now acceptable for publication, you may indicate that here to bypass the “Comments to the Author” section, enter your conflict of interest statement in the “Confidential to Editor” section, and submit your "Accept" recommendation.

Reviewer #3: (No Response)

Reviewer #5: All comments have been addressed

2. Is the manuscript technically sound, and do the data support the conclusions?

Reviewer #3: Yes

Reviewer #5: Yes

3. Has the statistical analysis been performed appropriately and rigorously?

Reviewer #3: No

Reviewer #5: Yes

4. Have the authors made all data underlying the findings in their manuscript fully available?

Reviewer #3: Yes

Reviewer #5: Yes

5. Is the manuscript presented in an intelligible fashion and written in standard English?

Reviewer #3: Yes

Reviewer #5: Yes

6. Review Comments to the Author

Reviewer #3: This is the second revision I have reviewed after I sent comments for revision. Incredibly, the authors have still not addressed my suggestions. For example, References #25 and 28 are still identical. The fact that "Group 4" only contains 1 member and yet is dealt with statistically, is not acceptable.

I have nothing further to say, and will not render any other decisions. I WILL NOT REVIEW THIS MANUSCRIPT AGAIN.

The changes made are: 

- group 4: indicated that it was not included in statistical analysis being a single case, the corresponding column from table 1 has been deleted

- reference number 28 has been deleted and the numbering within the text adjusted

Reviewer #5: All questions mentioned in previous review have received more detailed and serious responses.The writing of the article has also been thoroughly improved.

7. PLOS authors have the option to publish the peer review history of their article (what does this mean?). If published, this will include your full peer review and any attached files.

Do you want your identity to be public for this peer review? For information about this choice, including consent withdrawal, please see our Privacy Policy.

Reviewer #3: No

Reviewer #5: No

---

## [Decision Letter · Decision Letter 4]

1 Nov 2021

PONE-D-20-24720R4Margin between success and failure of PDA stenting for  duct-dependent pulmonary circulationPLOS ONE

Dear Dr. Agha,

Thank you for submitting your manuscript to PLOS ONE. After careful consideration, we feel that it has merit but does not fully meet PLOS ONE’s publication criteria as it currently stands. Therefore, we invite you to submit a revised version of the manuscript that addresses the points raised during the review process.

Please answer the comments by Reviewer #2.

We look forward to receiving your revised manuscript.

Kind regards,

Academic Editor

PLOS ONE

Journal Requirements:

Reviewers' comments:

Reviewer's Responses to Questions

**Comments to the Author**

1. If the authors have adequately addressed your comments raised in a previous round of review and you feel that this manuscript is now acceptable for publication, you may indicate that here to bypass the “Comments to the Author” section, enter your conflict of interest statement in the “Confidential to Editor” section, and submit your "Accept" recommendation.

Reviewer #2: (No Response)

Reviewer #5: All comments have been addressed

2. Is the manuscript technically sound, and do the data support the conclusions?

Reviewer #2: No

Reviewer #5: Yes

3. Has the statistical analysis been performed appropriately and rigorously? 

Reviewer #2: No

Reviewer #5: Yes

4. Have the authors made all data underlying the findings in their manuscript fully available?

Reviewer #2: Yes

Reviewer #5: Yes

5. Is the manuscript presented in an intelligible fashion and written in standard English?

Reviewer #2: Yes

Reviewer #5: Yes

6. Review Comments to the Author

Reviewer #2: This study reports on the use of PDA stenting in the setting of resource-constrained country where availabiity of surgica shunt is limited. Although of some interest this reviewer think that this study is not adding significant novelty to the field.

Reviewer #5: All questions mentioned in previous review have received more detailed responses. No more suggestions.

7. PLOS authors have the option to publish the peer review history of their article (what does this mean?). If published, this will include your full peer review and any attached files.

Reviewer #2: No

Reviewer #5: No

---

## [Author Response · Author response to Decision Letter 4]

24 Jan 2022

Rebuttal letter

The authors thank the editors and the reviewers for their valuable comments, this letter includes a response to the individual remarks

1. If the authors have adequately addressed your comments raised in a previous round of review and you feel that this manuscript is now acceptable for publication, you may indicate that here to bypass the “Comments to the Author” section, enter your conflict of interest statement in the “Confidential to Editor” section, and submit your "Accept" recommendation.

Reviewer #2: (No Response)

Reviewer #5: All comments have been addressed

Response: Thank you for your kind response 

2. Is the manuscript technically sound, and do the data support the conclusions?

Reviewer #2: No

Reviewer #5: Yes

Response: Thank you for your kind response. The conclusion was based upon the supported data and results of statistical significance 

3. Has the statistical analysis been performed appropriately and rigorously?

Reviewer #2: No

Reviewer #5: Yes

Response: Thank you for your kind response .

Numerical data were expressed as median and interquartile range [25th -75th percentiles] while categorical data were expressed as numbers or numbers and percentages. Comparisons between groups were calculated using non-parametric Kruskal Wallis, Mann-Whitney and Wilcoxon signed-rank tests for numerical data, while χ2 tests were used for comparison between groups for categorical data. Logistic regression was performed for evaluation of outcome predictors. Statistical significance was considered if P-values <0.05.

4. Have the authors made all data underlying the findings in their manuscript fully available?

Reviewer #2: Yes

Reviewer #5: Yes

Response: Thank you for your kind response 

5. Is the manuscript presented in an intelligible fashion and written in standard English?

Reviewer #2: Yes

Reviewer #5: Yes

Response: Thank your for you kind response 

6. Review Comments to the Author

Reviewer #2: This study reports on the use of PDA stenting in the setting of resource-constrained country where availabiity of surgica shunt is limited. Although of some interest this reviewer think that this study is not adding significant novelty to the field.

Reviewer #5: All questions mentioned in previous review have received more detailed responses. No more suggestions.

Response: Thank you for your kind response. Priority for ductal stenting will be in countries with limited resources, long waiting surgical lists and unavailability of small B-T shunt so the authors think about the treatment of duct dependent pulmonary circulation and different factors that predict the outcome of this procedure.

7. PLOS authors have the option to publish the peer review history of their article (what does this mean?). If published, this will include your full peer review and any attached files.

Do you want your identity to be public for this peer review? For information about this choice, including consent withdrawal, please see our Privacy Policy.

Reviewer #2: No

Reviewer #5: No

---

## [Decision Letter · Decision Letter 5]

23 Feb 2022

Margin between success and failure of PDA stenting for  duct-dependent pulmonary circulation

PONE-D-20-24720R5

Dear Dr. Agha,

We’re pleased to inform you that your manuscript has been judged scientifically suitable for publication and will be formally accepted for publication once it meets all outstanding technical requirements.

Kind regards,

Academic Editor

PLOS ONE

Additional Editor Comments (optional):

Reviewers' comments:

Reviewer's Responses to Questions

**Comments to the Author**

1. If the authors have adequately addressed your comments raised in a previous round of review and you feel that this manuscript is now acceptable for publication, you may indicate that here to bypass the “Comments to the Author” section, enter your conflict of interest statement in the “Confidential to Editor” section, and submit your "Accept" recommendation.

Reviewer #2: (No Response)

Reviewer #5: All comments have been addressed

2. Is the manuscript technically sound, and do the data support the conclusions?

Reviewer #2: No

Reviewer #5: Yes

3. Has the statistical analysis been performed appropriately and rigorously? 

Reviewer #2: No

Reviewer #5: Yes

4. Have the authors made all data underlying the findings in their manuscript fully available?

Reviewer #2: Yes

Reviewer #5: Yes

5. Is the manuscript presented in an intelligible fashion and written in standard English?

Reviewer #2: No

Reviewer #5: Yes

6. Review Comments to the Author

Reviewer #2: This manuscript has been revised and this reviewer recognize the effort by these authors. However, I still continue to think that procedural indication is unclear, main study results are not novel, in addition suggesting PDA stenting over shunt is a delicate and unresolved matter in managing neonates with ductal-dependant pulmonary circulation and this study does not offer any insights to this regard.

Reviewer #5: No more suggestions.

All comments have been addressed carefully.

In the case of limited resources, catheter stenting is a hope for research subjects, and refining the prognosis and characteristics of catheter-dependent pulmonary circulation stenting can provide more patients with accurate and effective interventions

7. PLOS authors have the option to publish the peer review history of their article (what does this mean?). If published, this will include your full peer review and any attached files.

Reviewer #2: No

Reviewer #5: No

---

## [Editor Report · Acceptance letter]

25 Mar 2022

PONE-D-20-24720R5 

Margin between success and failure of PDA stenting for duct-dependent pulmonary circulation 

Dear Dr. Agha:

I'm pleased to inform you that your manuscript has been deemed suitable for publication in PLOS ONE. Congratulations! Your manuscript is now with our production department. 

Kind regards, 

on behalf of

Dr. Robert Jeenchen Chen 

Academic Editor

PLOS ONE